# Phenotypic changes of HER2-positive breast cancer during and after dual HER2 blockade

Fara Brasó-Maristany[1,2], Gaia Griguolo[1,2,3,4], Tomás Pascual [1,2,5], Laia Paré[5], Paolo Nuciforo [6,7], Antonio Llombart-Cussac[8], Begoña Bermejo[9], Mafalda Oliveira [6,7], Serafín Morales[10], Noelia Martínez[11], Maria Vidal[1,2,5], Barbara Adamo[1,2], Olga Martínez [1,2], Sonia Pernas[5,12], Rafael López[13], Montserrat Muñoz[1,2], Núria Chic[1,2], Patricia Galván[1,2], Isabel Garau[14], Luis Manso[15], Jesús Alarcón[16], Eduardo Martínez[17], Sara Gregorio[18], Roger R. Gomis [18], Patricia Villagrasa[5], Javier Cortés[7,19], Eva Ciruelos[5,15] & Aleix Prat[1,2,5]*

The HER2-enriched (HER2-E) subtype within HER2-positive (HER2+) breast cancer is highly addicted to the HER2 pathway. However, ~20–60% of HER2+/HER2-E tumors do not achieve a complete response following anti-HER2 therapies. Here we evaluate gene expression data before, during and after neoadjuvant treatment with lapatinib and trastuzumab in HER2+/HER2-E tumors of the PAMELA trial and breast cancer cell lines. Our results reveal that dual HER2 blockade in HER2-E disease induces a low-proliferative Luminal A phenotype both in patient's tumors and in vitro models. These biological changes are more evident in hormone receptor-positive (HR+) disease compared to HR-negative disease. Interestingly, increasing the luminal phenotype with anti-HER2 therapy increased sensitivity to CDK4/6 inhibition. Finally, discontinuation of HER2-targeted therapy in vitro, or acquired resistance to anti-HER2 therapy, leads to restoration of the original HER2-E phenotype. Our findings support the use of maintenance anti-HER2 therapy and the therapeutic exploitation of subtype switching with CDK4/6 inhibition.

[1] Department of Medical Oncology, Hospital Clínic de Barcelona, Carrer de Villarroel, 170, 08036 Barcelona, Spain. [2] Translational Genomics and Targeted Therapeutics in Solid Tumors, August Pi i Sunyer Biomedical Research Institute (IDIBAPS), Carrer del Rosselló, 149-153, 08036 Barcelona, Spain. [3] Department of Surgery, Oncology and Gastroenterology, University of Padova, Via Giustiniani, 2, 35124 Padova, Italy. [4] Medical Oncology 2, Istituto Oncologico Veneto IRCCS, Via Gattamelata, 64, 35128 Padova, Italy. [5] SOLTI Breast Cancer Research Group, Carrer de Balmes, 115, 08008 Barcelona, Spain. [6] Vall d'Hebrón University Hospital, Passeig de la Vall d'Hebron, 119-129, 08035 Barcelona, Spain. [7] Vall d´Hebron Institute of Oncology (VHIO), Carrer de Natzaret, 115-117, 08035 Barcelona, Spain. [8] Hospital Universitario Arnau de Vilanova, Carrer de Sant Clement, 12, 46015 Valencia, Spain. [9] Hospital Clínico Universitario de Valencia, Av. de Blasco Ibáñez, 17, 46010 Valencia, Spain. [10] Hospital Universitario Arnau de Vilanova, Av. Alcalde Rovira Roure, 80, 25198 Lleida, Spain. [11] Hospital Universitario Ramón y Cajal, M-607, km. 9, 100, 28034 Madrid, Spain. [12] Institut Catala d'Oncologia, Avinguda de la Gran Via de l'Hospitalet, 199-203, 08908 Hospitalet de Llobregat, Spain. [13] Hospital Clínico Universitario de Santiago, Rúa da Choupana, s/n, 15706 Santiago de Compostela, Spain. [14] Hospital Son Llàtzer, Ctra. de Manacor, 07198 Palma de Mallorca, Spain. [15] Hospital Universitario 12 de Octubre, Av. de Córdoba, s/n, 28041 Madrid, Spain. [16] Hospital Universitario Son Espases, Carretera de Valldemossa, 79, 07120 Palma de Mallorca, Spain. [17] Consorcio Hospitalario Provincial de Castellón, Av. del Dr. Clarà, 19, 12002 Castellón de la Plana, Spain. [18] Institute for Research in Biomedicine, Carrer de Baldiri Reixac, 10, 08028 Barcelona, Spain. [19] IOB Institute of Oncology, Quiron Group, Plaça d'Alfonso Comín, 5, 08023 Barcelona, Spain. *email: alprat@clinic.cat

Breast cancer consists of four molecular intrinsic subtypes (i.e., Luminal A, Luminal B, HER2-enriched [HER2-E] and Basal-like) and a normal-like group[1–5]. Among them, the HER2-E is characterized by high expression of growth factor receptor-related genes (*ERBB2, EGFR* and/or *FGFR4*) and cell cycle-related genes, low expression of estrogen-related genes such as *ESR1* and *PGR*, and low expression of basal-related genes[6]. Although the HER2-E subtype is related to HER2+ disease as defined by immunohistochemistry, important discrepancies exist and HER2-E subtype represents ∼60%, ∼80% and ∼40% of HER2+, HER2+/hormone receptor-negative (HER2+/HR-negative) and HER2+/HR+ tumors, respectively[7].

The clinical value of the HER2-E subtype in HER2+ breast cancer is starting to be elucidated. From a therapeutic perspective, the HER2-E subtype has shown, across 14 clinical trials and ∼2,000 patients, higher sensitivity to anti-HER2-based therapies than the non-HER2-E subtypes[7–20]. However, not all HER2-E tumors achieve a complete response following anti-HER2-based therapies. For example, the pathological complete response (pCR) rates in HER2-E subtype following anti-HER2-based neoadjuvant therapy, with or without chemotherapy, are 40–80%[21]. At the same time, HER2-E tumors that do not achieve a pCR have a poor survival outcome[22]. Thus, there is a need to better understand the biology associated with incomplete response to HER2-targeted therapy in HER2-E disease. Here, we study patient's tumor samples and preclinical models before, during and after anti-HER2 therapy to shed light on this clinical observation.

## Results

### Early in vivo biological changes during dual HER2 blockade.
To identify early molecular changes induced by dual HER2 blockade in patients with HER2-E disease, gene expression profiling was performed in 96 tumor samples obtained before and at day 14 of treatment with lapatinib and trastuzumab (and endocrine therapy if the tumor was HR+) in the PAMELA phase II clinical trial[7]. The expression of the 50 PAM50 genes and 6 signatures (Basal-like, HER2-E, Luminal A, Luminal B, normal-like and the 11-gene proliferation score) was explored at both time-points (Fig. 1a). Among the 56 variables, 85.7% and 94.7% were found differentially expressed (False Discovery Rate [FDR]<5%) in HR+ ($n = 35$) and HR-negative ($n = 61$) disease, respectively (Supplementary Table 1). The magnitude of expression of the transcriptional profiles in HER2+/HR+/HER2-E and HER2+/HR-negative/HER2-E disease was strongly correlated (Pearson correlation coefficient = 0.93, *P*-value ($p$) < 0.001) (Supplementary Fig. 1a). Overall, a significant relative increase in Luminal A and normal-like signature scores, and a relative decrease in proliferation, HER2-E and Luminal B scores, was observed at day 14 compared to baseline (Fig. 1b). Although similar biological changes were observed in HER2+/HR+/HER2-E and HER2+/HR-negative/HER2-E diseases, a PAM50 subtype switch from HER2-E to Luminal A was observed in 31.6% of HR+ tumors and 4.8% of HR-negative tumors (Fig. 1c). The observed shift to Luminal A subtype was due to a decrease in cell proliferation-related genes and a significant increase in the expression of luminal-related genes. Concordant with this finding, Ki67 positivity by immunohistochemistry, determined in tumor cells at day 14 (Supplementary Fig. 1b), was lower in PAM50 Luminal A tumor samples compared to PAM50 non-Luminal A tumor samples (mean Ki67 of 10.8% vs. 19.8%; $p = 0.009$ by two-tailed unpaired *t*-test). Of note, normal tissue contamination at day 14 did not differ between PAM50 Luminal A tumor samples compared to PAM50 non-Luminal A tumor samples (mean tumor cellularity of 39.8% vs. 41.2%; $p = 0.839$ by two-tailed unpaired *t*-test) (Supplementary Fig. 1c). Finally, since

patients with HER2+/HR+ disease in the PAMELA trial received anti-HER2 therapy in combination with letrozole or tamoxifen, and treatment with letrozole alone for 2 weeks has shown to reduce Ki67 in HER2+/HR+ tumors in the PER-ELISA phase II trial, specially within Luminal A and B disease[10], we separated and compared the gene expression changes at day 14 in the PAMELA trial of HER2+/HR+/Luminal A or B tumors ($n = 38$) with HER2+/HR+/HER2-E tumors ($n = 38$). The analysis revealed a strong correlation (Pearson correlation coefficient = 0.89, $p < 0.001$) between both gene lists, suggesting that anti-HER2 therapies have a higher impact on molecular tumor changes at day 14 than endocrine therapy (Supplementary Fig. 1d), although we cannot exclude an additive or a synergistic effect between anti-HER2 therapy and endocrine therapy in this group of patients.

Next, we assessed the expression of the 56 genes/signatures in paired tumor samples from 36 patients with HER2+/HER2-E disease recruited in the LPT109096 phase II clinical trial[23], where patients were treated for 2 weeks with either lapatinib, trastuzumab or the combination. Among the 56 genes/signatures, 85.7% were found differentially expressed (Supplementary Table 2). As expected, the gene expression profiles of the LPT109096 and PAMELA trials were strongly correlated (Pearson correlation coefficient = 0.87, $p < 0.001$) (Supplementary Fig. 2a). When each single arm was evaluated, no major differences were observed, although lapatinib seems to have a greater impact on molecular changes than trastuzumab (Supplementary Fig. 2b–e). However, the small sample size within each arm did not allow formal statistical comparison. Of note, intrinsic subtype switching was not observed in tumor samples from the LPT109096 study, possibly due to the small sample size of the combination arm ($n = 8$) or the lack of endocrine therapy.

### Early in vitro biological changes during dual HER2 blockade.
To further evaluate the effects of dual HER2 blockade, we performed several experiments in 3 HER2+ breast cancer cell lines (BT474, which is HR+, SKBR3 and HCC1954, which are HR-negative) and 1 HER2-negative cell line (MCF7, which is HR+). PAM50 subtyping of the 3 cell lines revealed that BT474, SKBR3 and HCC1954 are HER2-E and MCF7 is Luminal B (Supplementary Fig. 3a). These in vitro models were treated with trastuzumab, lapatinib (an EGFR/HER2 tyrosine kinase inhibitor [TKI]), neratinib (a pan-HER TKI) or tucatinib (a HER2 TKI) as single agents (Supplementary Fig. 3b, c) or in combination. As expected, all anti-HER2 treatments evaluated reduced cell viability in a dose-dependent manner in BT474 and SKBR3 (Fig. 2a and Supplementary Fig. 3b, c) but not in HCC1954, which has a mutation in *PIK3CA* that might confer resistance to anti-HER2 therapy[24], or MCF7, which does not overexpress HER2 (Supplementary Fig. 3d). Concordant with this observation, phosphorylation of HER2 and phosphorylation of the survival kinase AKT were reduced upon treatment in BT474 and SKBR3, demonstrating successful pathway inhibition (Fig. 2b). Dual HER2 blockade arrested cell cycle at G1 (Supplementary Fig. 4a) and reduced clonogenic potential (Supplementary Fig. 4b), consistent with known roles of HER2 downstream signaling pathways[25].

Next, we evaluated the PAM50 subtypes, genes and signature scores in BT474 and SKBR3 cells before and after 72 h treatment with different combinations of trastuzumab (10 μg ml$^{-1}$) and a TKI (given at the IC50 for each cell line). Regarding individual PAM50 genes and signatures, 100% and 92.8% were found differentially expressed in BT474 and SKBR3, respectively (Supplementary Table 3), and the magnitudes of gene expression profiles were strongly correlated between the two cell lines

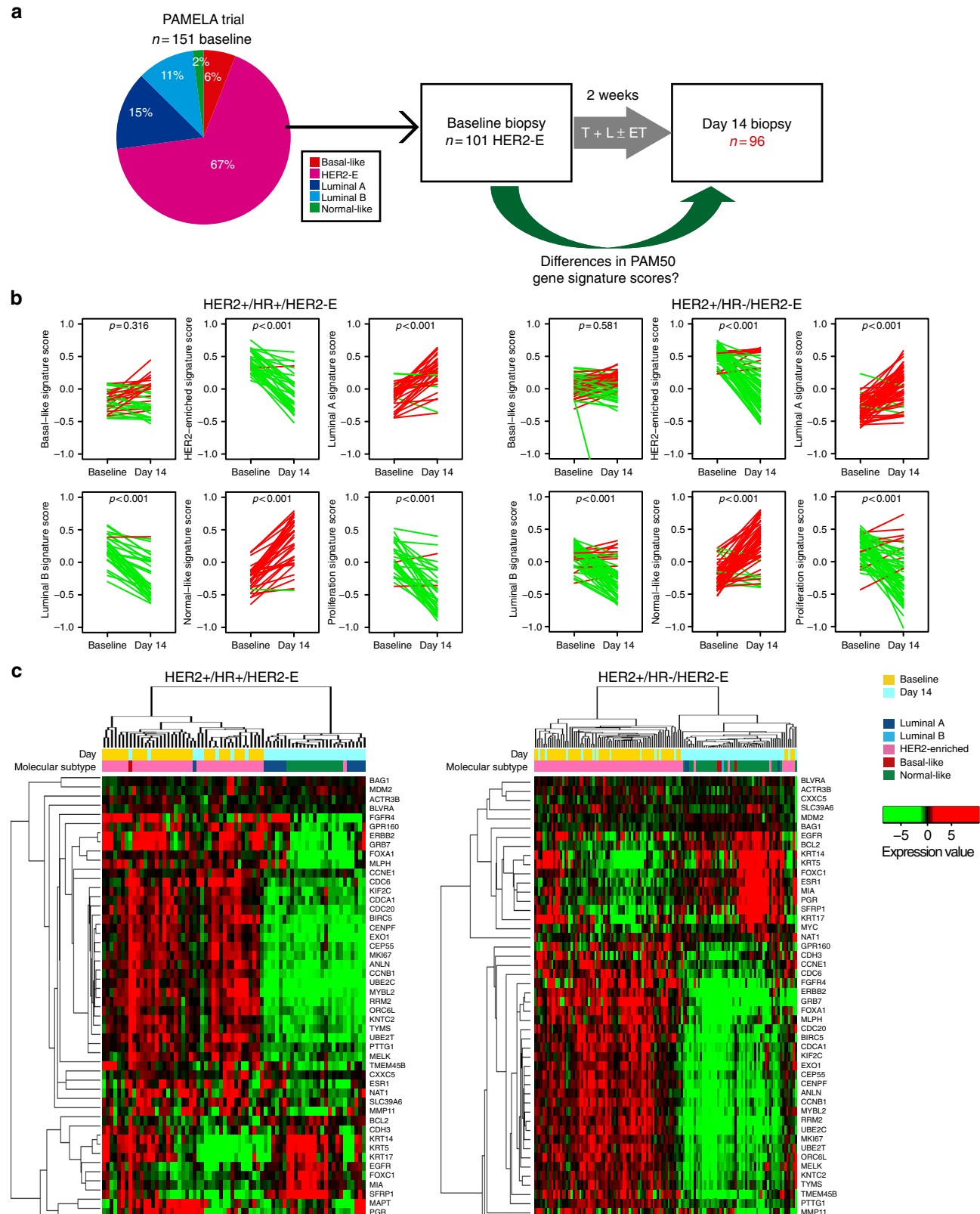

**Fig. 1 HER2 blockade led to a Luminal A phenotype in HER2-E tumors. a** Schematic representation of the workflow to identify differences in PAM50 gene expression signatures in HER2-E tumors (*n* = 151) of the PAMELA trial between baseline and day 14 of treatment with lapatinib (L) plus trastuzumab (T) plus endocrine therapy (ET) if HR+. **b** PAM50 signature scores in HER2+/HR+/HER2-E at baseline (left) and HER2+/HR-negative/HER2-E at baseline (right) tumors treated with dual HER2 blockade at baseline and day 14. Each line represents a tumor sample. Increases are represented in red and decreases in green. *P*-values (*p*) in **a** and **b** were determined by two-tailed paired *t*-tests. **c** Unsupervised hierarchical clustering across 35 paired HER2+/HR+/HER2-E at baseline tumors and 61 paired HER2+/HR-negative/HER2-E at baseline tumors. Heatmaps show high (red) to low (green) expression of mRNAs in each sample. The day of biopsy and molecular subtype calls of each sample are shown. Source data are provided as a Source Data file.

(Pearson correlation coefficient = 0.75, $p < 0.001$) and the gene expression profile of each cell line was moderately correlated (Pearson correlation coefficients = 0.46–0.76, $p < 0.001$) with the gene expression profiles obtained from PAMELA and LPT109096 (Supplementary Fig. 5a, b). Similar to patient's tumor samples, dual HER2 blockade in cell lines led to a significant relative increase in Luminal A and normal-like signature scores, and a relative decrease in proliferation, HER2-E, Luminal B and Basal-like signature scores (Fig. 2c). Regarding PAM50 subtype changes, a switch to Luminal A following dual HER2 blockade was observed in BT474 (HER2+/HR+/HER2-E) but not in SKBR3 (HER2+/HR-negative/HER2-E) (Fig. 2d). Overall, the biological changes observed in our in vitro models recapitulated the biological changes observed in patients with HER2-E tumors treated with dual HER2 blockade.

**Biological changes after dual HER2 blockade.** We then sought to better understand the biology of HER2-E tumors after treatment with dual HER2 blockade. To approach this, we completed PAM50 gene expression analysis in residual tumors of 57 patients with HER2-E recruited in the PAMELA who received 18 weeks of dual HER2 blockade (Fig. 3a). Of note, the median time from the last dose of therapy to surgery, where the tumor samples were obtained, was 29.5 days (range = 7–76 days). Among the 56 PAM50 genes and signatures, 92.8% were found differentially expressed in residual tumors at surgery compared to day 14 (Supplementary Table 4). The vast majority of genes or signatures (46/56, 82.1%) were found up-regulated at surgery compared to day 14, including all proliferation-related genes (i.e., *UBE2T*, *EXO1*, *CDCA1*, *TYMS*, *CEP55*, *CDC20*, *UBE2C*, or *MKI67*) and the proliferation signature (Fig. 3b). Among the down-regulated genes or signatures (10/56, 18%), we identified the Luminal A signature (Fig. 3b). However, we did not observe a correlation between the rebound effect and the number of days from the last dose and the day of the surgical procedure (Supplementary Fig. 6).

Following these observations in patient's tumors, we hypothesized that discontinuation of dual HER2 inhibition reverts its biological effects. Therefore, we evaluated the biological effects in preclinical models after interrupting anti-HER2 treatment. BT474 and SKBR3 cells were treated with combinations of trastuzumab and a single TKI for 72 h, treatment was then discontinued and cells were cultured for another 72 h without treatment. Gene expression analysis revealed a rebound effect in 75% and 100% PAM50 genes and signatures in BT474 and SKBR3 respectively after anti-HER2 therapy discontinuation (Supplementary Table 5). We observed a significant relative decrease in Luminal A and normal-like signature scores, and a significant relative increase in Proliferation, Luminal B and Basal-like signature scores in BT474 and SKBR3 cells (Fig. 3c). The gene expression profiles of both cell lines were strongly correlated (Pearson correlation coefficient = 0.69, $p < 0.001$). Finally, BT474 cell line was identified by PAM50 as HER2-E as the parental cell line once anti-HER2 therapy was withdrawn (Fig. 3d).

**Biological changes during anti-HER2 resistance.** To understand the changes in molecular phenotypes induced by chronic HER2 inhibition, we established a BT474-derived lapatinib and trastuzumab resistant (BT474-L$^R$T$^R$) cell line and a BT474-derived tucatinib and trastuzumab resistant (BT474-Tu$^R$T$^R$) cell line. Cell viability assays demonstrated that BT474-L$^R$T$^R$ and BT474-Tu$^R$T$^R$ cells were resistant to combinations of lapatinib plus trastuzumab, or tucatinib plus trastuzumab, as opposed to parental BT474 (Fig. 4a). BT474-L$^R$T$^R$ and BT474-Tu$^R$T$^R$ were cultured with 2 µg ml$^{-1}$ trastuzumab and 2 nM of lapatinib or tucatinib, respectively, concentrations at which the parental

BT474 became Luminal A. However, BT474-L$^R$T$^R$ and BT474-Tu$^R$T$^R$ remained HER2-E (Fig. 4b). We subsequently compared the molecular profile of BT474-L$^R$T$^R$ and BT474-Tu$^R$T$^R$ cell lines with their parental HER2-E BT4T4 cell line. Among the 56 variables, 39.9% and 16.1% were significantly upregulated and 60.7% and 44.7% were significantly downregulated in BT474-L$^R$T$^R$ and BT474-Tu$^R$T$^R$ respectively. Both transcriptional profiles were highly correlated (Pearson correlation coefficient = 0.83, $p < 0.001$), with FGFR4 being the most upregulated gene (Supplementary Table 6). Interestingly, FGFR4 is a known driver of the HER2-E phenotype[26]. Finally, no differences were observed between both resistant cell lines (i.e., BT474-L$^R$T$^R$ and BT474-Tu$^R$T$^R$).

**Dual HER2 inhibition and sensitivity to CDK4/6 inhibition.** Whether the shift to the Luminal A phenotype induced by dual HER2 inhibition could sensitize HER2-E cell lines to highly effective therapies in patients with luminal disease, such as CDK4/6 inhibitors[27–29], is unknown. To approach this, BT474 and SKBR3 cells were treated for 24 h with either trastuzumab or a single TKI to induce a Luminal A phenotype. HCC1954 was used as a negative control, as it is intrinsically resistant to anti-HER2 treatments. Palbociclib was then added in combination with an anti-HER2 treatment for 72 h. Surprisingly, BT474 and SKBR3 cells became more sensitive to palbociclib after being exposed to anti-HER2 therapies, indicating that the acquisition of a more luminal-like phenotype can sensitize cells to CDK4/6 inhibition. As expected, anti-HER2 treatments did not increase sensitivity to palbociclib in HCC1954. At the same time, both BT474-L$^R$T$^R$ and BT474-Tu$^R$T$^R$ cell lines were insensitive to palbociclib (Fig. 5a), further demonstrating that the HER2-E phenotype is resistant to anti-CDK4/6 treatments. Overall, this data suggested that anti-HER2 treatment can modulate the sensitivity to CKD4/6 inhibition in HER2-E disease by inducing a Luminal A-like phenotype, while cells that remain HER2-E (such as the BT474-derived HER2-resistant cells) do not respond to these treatments (Fig. 5b). As expected, anti-HER2 therapy reduced the expression levels of Cyclin D1 and the phosphorylation of retinoblastoma (RB) and the combination of HER2 and CDK4/6 inhibitors reduced the phosphorylation of RB more than CDK4/6 inhibitor alone (Fig. 5c). Concordant with this observation, dual HER2 blockade led to a decrease in *CCND1* mRNA levels at week 2 in HER2+/HR+ disease of the PAMELA trial (Fig. 5d). Altogether, these data suggest that the combination is more efficient at arresting cell cycle and preventing tumor growth (Fig. 5e).

## Discussion
The HER2-E intrinsic molecular subtype has higher anti-HER2-sensitivity compared to other molecular subtypes[7–20]. However, the molecular and phenotypic characteristics of the HER2+/HER2-E resistant tumors are not well understood. This is critical, as 20–30% and 50–70% of patients with HER2+/HER2-E tumors do not achieve a pCR following dual HER2 blockade with and without chemotherapy, respectively.

Our main observation is that HER2-E tumors cells that are sensitive to anti-HER2 therapy but do not die acquire a Luminal A phenotype. This is especially relevant in HER2+/HR+ disease. According to our results, the acquisition of this phenotype is relatively rapid (i.e., 14 days in tumors and 72 h in cell lines) and leads to anti-HER2 resistance. For example, HER2-E tumors that became Luminal A at day 14 in the PAMELA trial[7] had a 20% pCR rate upon completion of the neoadjuvant treatment compared to 55.8% in those HER2-E tumors that became normal-like, a biomarker of tumor responsiveness and stromal contamination. This switch into a Luminal A phenotype has also been reported following neoadjuvant

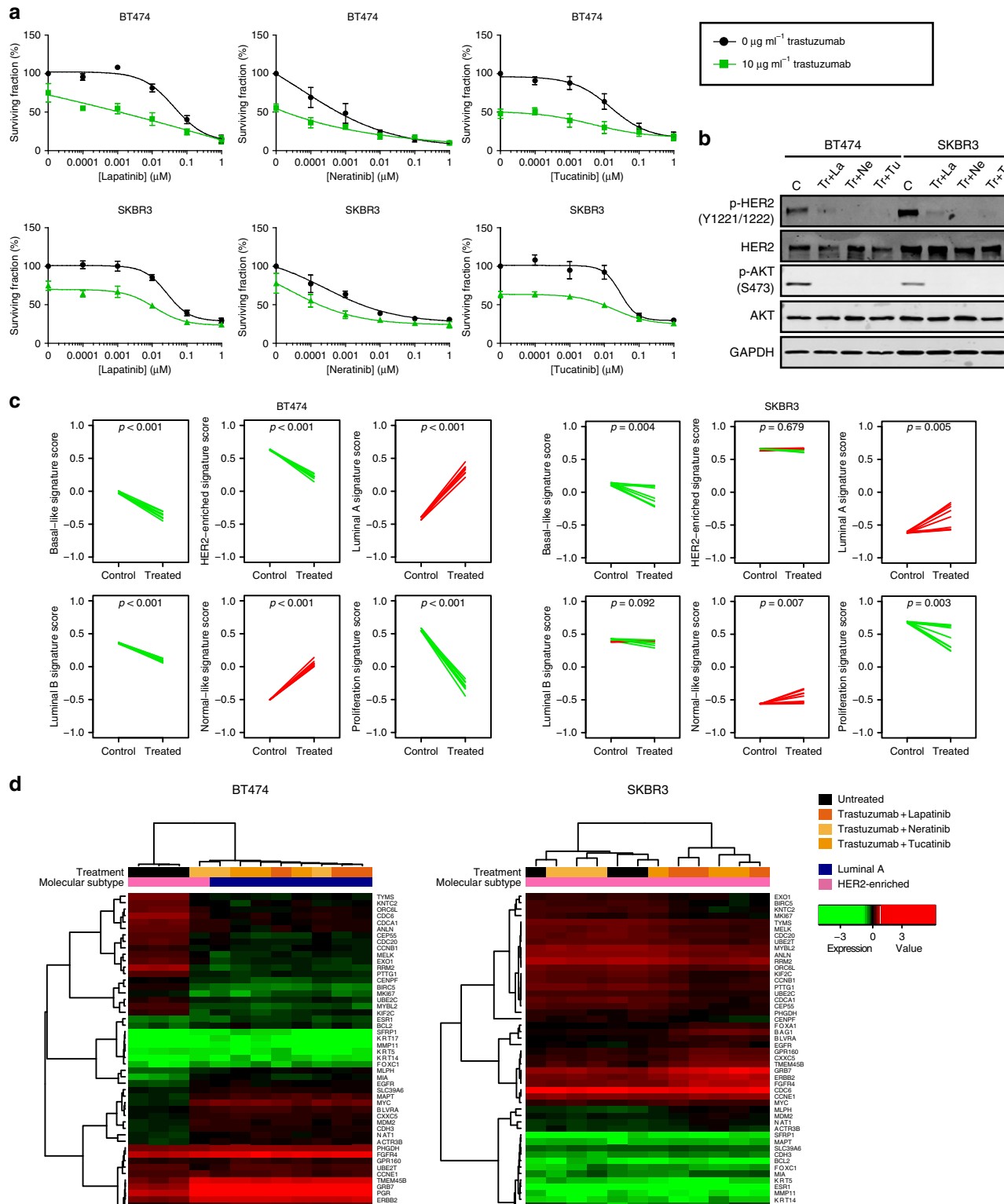

**Fig. 2 Effects of anti-HER2 treatments in HER2-E breast cancer cell lines. a** Cell viability (%) of BT474 and SKBR3 cells upon treatment with increasing concentrations of the TKI lapatinib, neratinib or tucatinib as monotherapy or in combination with 10 μg ml$^{-1}$ trastuzumab for 72 h. Data points represent the mean; error bars represent the standard error of the mean of 3 independent experiments. **b** Phosphorylation and total levels of HER2 and AKT upon 24 h of treatment with 10 μg ml$^{-1}$ trastuzumab plus TKI (10 nM lapatinib, 10 nM neratinib, 10 nM tucatinib) as assessed by Western Blot. **c** PAM50 signature scores in BT474 and SKBR3 cells untreated and treated with combinations of TKI and trastuzumab for 72 h. Each line represents a paired sample. Increases are represented in red and decreases in green. *P*-values (*p*) were determined by two-tailed paired *t*-tests. **d** Unsupervised hierarchical clustering using the PAM50 genes across BT474 and SKBR3 cells treated with combinations of TKI and trastuzumab for 72 h. The heatmaps show high (red) to low (green) expression of mRNAs in each sample. The molecular subtype call and treatment of each sample is shown. Source data are provided as a Source Data file.

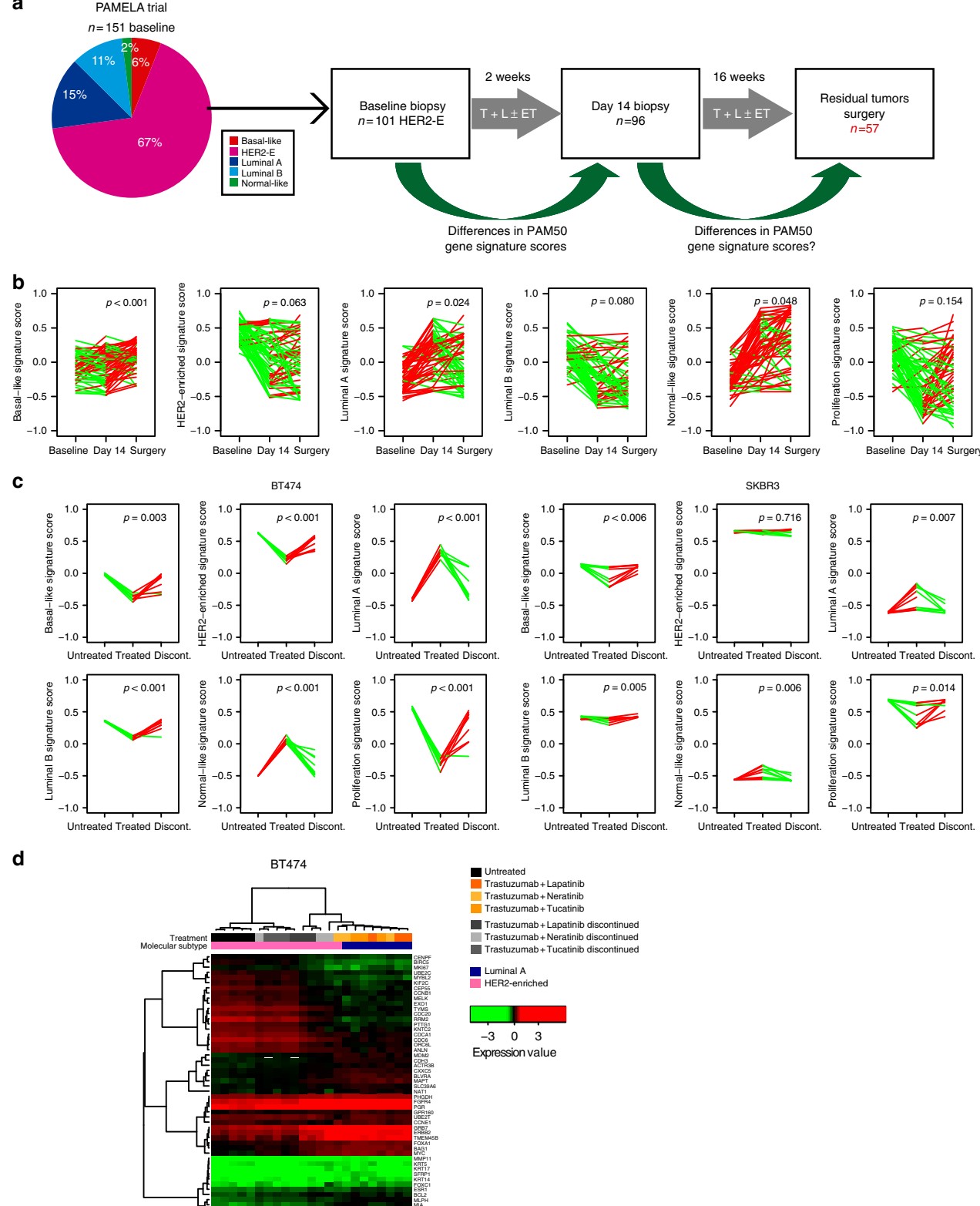

**Fig. 3 Biological changes after dual HER2 blockade. a** Schematic representation of the workflow to identify differences in PAM50 gene expression signatures in HER2-E tumors at baseline of the PAMELA trial between day 14 of treatment and residual tumors. **b** PAM50 signature expression changes between baseline, day 14 and surgery in 57 residual tumors at surgery. Each line represents a tumor sample. Increases are represented in red and decreases in green. *P*-values (*p*) were determined by two-tailed paired *t*-tests. **c** PAM50 signature scores in BT474 and SKBR3 cells untreated, treated with TKI + trastuzumab for 72 h and upon treatment discontinuation for 72 h. Each line represents a paired sample. *P*-values (*p*) were determined by two-tailed paired *t*-tests. **d** Unsupervised hierarchical clustering using the PAM50 genes across untreated BT474 cells, treated cells with combinations of TKI and trastuzumab for 72 h and cells where treatment was removed for another 72 h. The heatmap shows high (red) to low (green) expression of mRNAs in each sample. The molecular subtype call of each sample is shown.

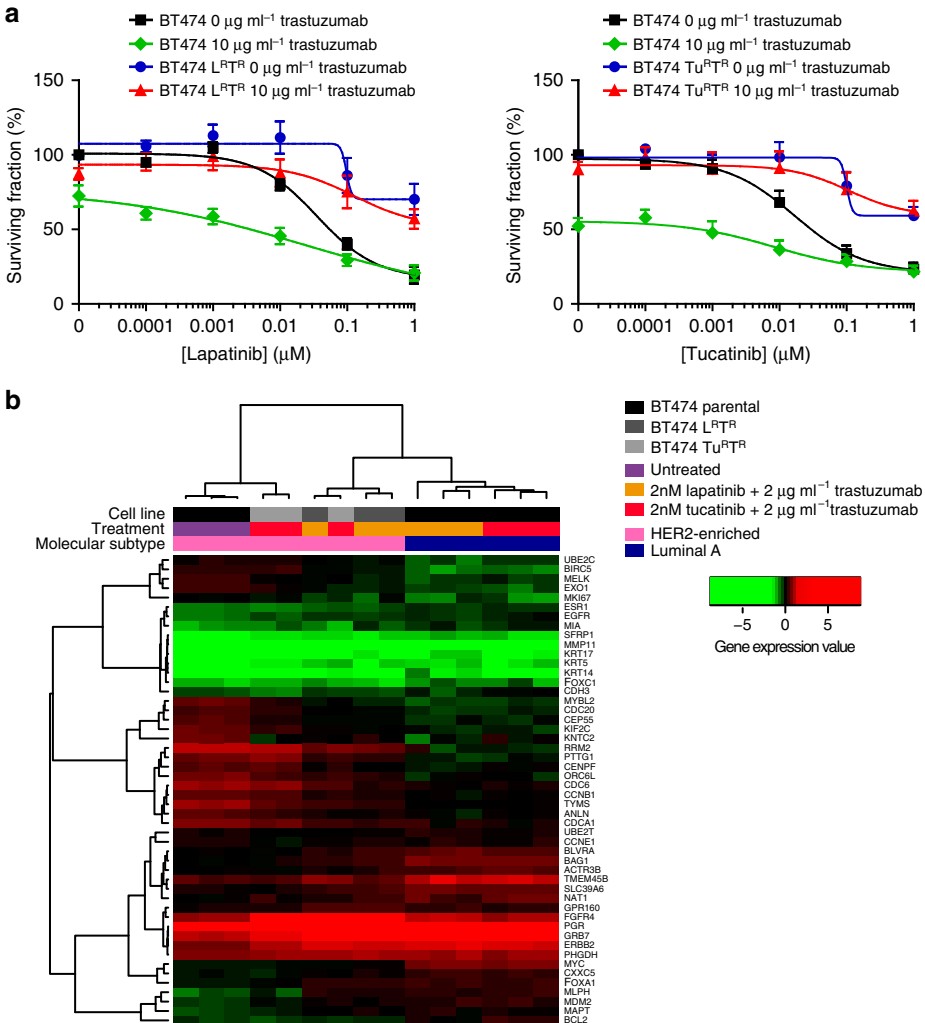

**Fig. 4 Biological changes during anti-HER2 resistance. a** Cell viability of BT474, BT474-derived lapatinib and trastuzumab resistant (BT474-L$^R$T$^R$) and BT474-derived tucatinib and trastuzumab resistant (BT474-Tu$^R$T$^R$) cells upon treatment with increasing doses of lapatinib +/− 10 μg ml$^{-1}$ trastuzumab or increasing doses of tucatinib +/− 10 μg ml$^{-1}$ trastuzumab, respectively. Data points represent the mean; error bars represent the standard error of the mean of 3 independent experiments. (**b**) Unsupervised hierarchical clustering in BT474, BT474-L$^R$T$^R$ and BT474-Tu$^R$T$^R$ treated with of 2 nM lapatinib plus 2 μg ml$^{-1}$ trastuzumab or 2 nM tucatinib plus 2 μg ml$^{-1}$ trastuzumab The heatmap shows high (red) to low (green) expression of mRNAs in each sample. The molecular subtype call and treatment of each sample is shown. Source data are provided as a Source Data file.

anti-HER2-based chemotherapy. In CALGB40601[13], NeoSphere[30] and NSABP B-41[20] trials, 30–67% of HER2-E tumors became Luminal A in residual disease. Not achieving a pCR after neoadjuvant treatment is associated with poor event free survival and overall survival in HER2+ breast cancer[31]. Altogether, these results identify the Luminal A phenotype as a predictive biomarker of resistance to anti-HER2 therapy.

Our study highlights two additional aspects. First, the subtype switch seems reversible upon stopping anti-HER2 therapy. This could explain why maintenance therapy with anti-HER2 therapy both in the adjuvant and metastatic settings are beneficial to patients, especially in high-risk patients[32]. Interestingly, a rebound effect after stopping treatment has also been seen with other therapies such as CDK4/6 inhibitors[28]. Second, subtype switching from HER2-E to Luminal A might open an opportunity to treat the acquired phenotype with drugs that are known to be active in Luminal A, such as endocrine therapy and/or CDK4/6 inhibitors. For example, the PATRICIA phase II trial[29] in advanced HER2+/HR+ disease revealed that the Luminal A tumors had improved progression-free survival when treated with palbociclib and trastuzumab as compared to HER2-E tumors. It is

important to note that patients in the PATRICIA trial had been previously treated with a mean of 3 prior lines of anti-HER2-based therapy. Thus, HER2-E tumors at this stage are likely to have high anti-HER2 resistance and switching to a Luminal A phenotype is unlikely. Although this is a hypothesis and needs further validation, our preclinical data using HER2+ cell lines with a complete resistance to anti-HER2 treatment supports this observation.

The HER2 signaling pathway directly regulates the expression of Cyclin D1, affecting its interaction with CDK4[33], this provides a rationale for the concomitant HER2 and CDK4/6 inhibition in patients. Indeed, combinations of anti-HER2 therapies with CDK4/6 inhibitor and endocrine therapy are being investigated in 2 phase III trials for advanced HER2+/HR+ disease (i.e., PATINA[34] and monarcHER[35]). Both trials differ in the population being evaluated. PATINA focuses on maintenance therapy in HER2+ tumors that are responding to anti-HER2 therapy, whereas monarcHER[35] focuses on tumors that have progressed to at least 2 prior lines of anti-HER2 therapy. Concordant with our hypothesis, the monarcHER main results presented at ESMO 2019[36] revealed that abemaciclib + fulvestrant + trastuzumab was

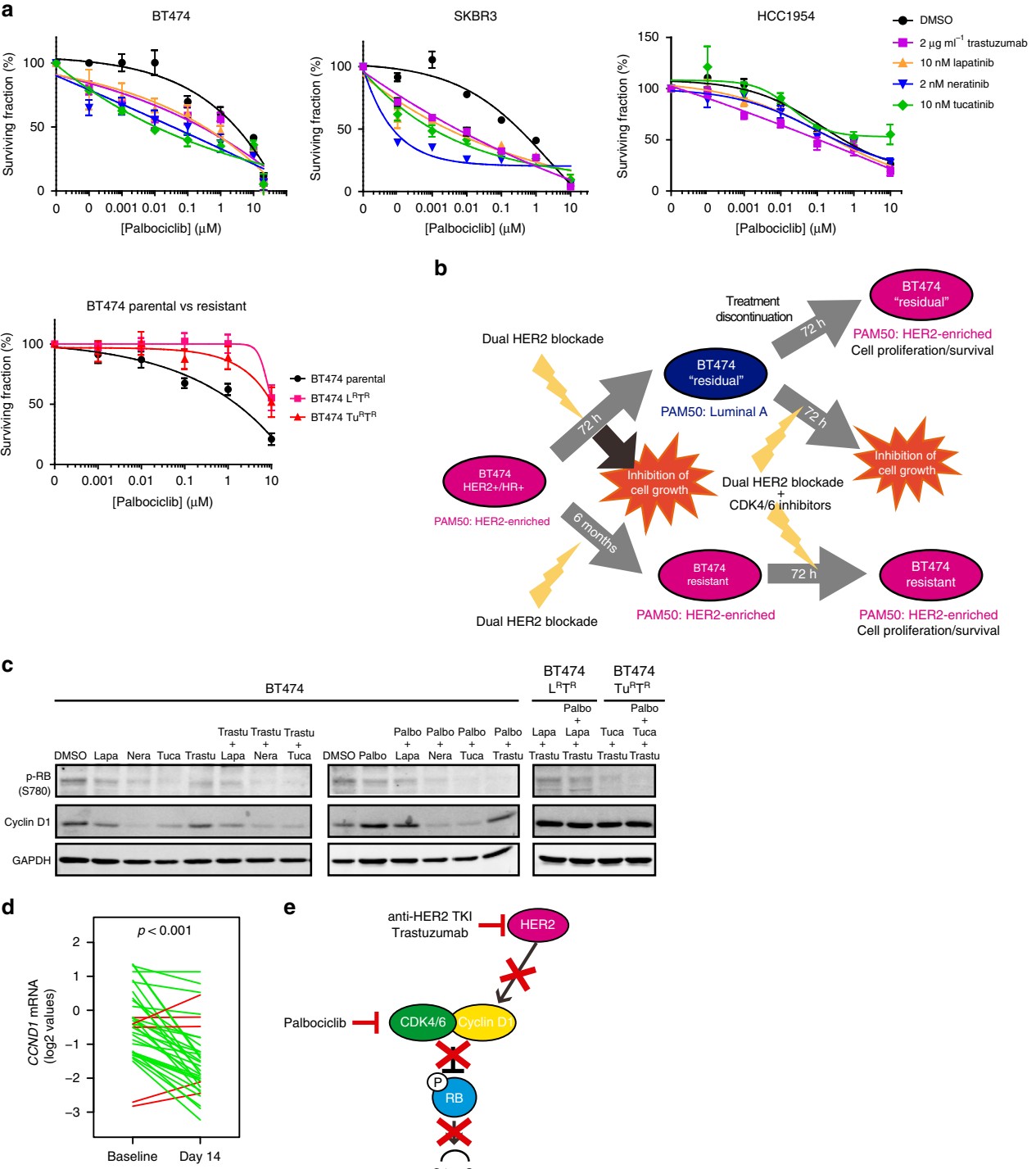

**Fig. 5 Shift to Luminal A sensitizes cells to anti-CDK4/6 treatments. a** Dose–response curve of BT474, SKBR3, HCC1954, BT474-derived lapatinib and trastuzumab resistant (BT474-L$^R$T$^R$) and BT474-derived tucatinib and trastuzumab resistant (BT474-Tu$^R$T$^R$) cells upon treatment with anti-HER2 drugs for 24 h and subsequent treatment with anti-HER2 plus increasing concentrations of palbociclib for 72 h. Surviving fraction was assessed by staining with Hoechst 33342. Data points represent the mean; error bars represent the standard error of the mean of 3 independent experiments. **b** Schematic representation of HER2-E breast cancer cells that are sensitive to dual HER2 blockade treatment, which triggers cell growth inhibition but also a shift to a Luminal A molecular phenotype in residual cells. These changes are reversible after treatment discontinuation, while inhibition of HER2 improves sensitivity to CDK4/6 targeted therapies. BT474 resistant to anti-HER2 treatments remain HER2-E and do not respond to inhibition of HER2 and CDK4/6. **c** Western Blot assessing the phosphorylation of RB and the expression of Cyclin D1 in BT474 cells upon 24 h of treatment with 10 nM TKI (lapatinib, neratinib, tucatinib), with 10 μg ml$^{-1}$ trastuzumab, with the combination of TKI plus trastuzumab, 10 nM palbociclib or with the combination of palbociclib with anti-HER2 treatments, and in BT474-L$^R$T$^R$ and BT474-Tu$^R$T$^R$ treated with 2 μg ml$^{-1}$ trastuzumab + 2 nM TKI (lapatinib and tucatinib respectively) with or without palbociclib. (**d**) *CCND1* mRNA levels in HER2+/HR+ tumors of the PAMELA trial at baseline and day 14. Each line represents a paired sample. Increases are represented in red and decreases in green. *P*-value (*p*) was determined by two-tailed paired *t*-tests. (**e**) Schematics of the mechanism by which double-targeting the HER2 and CDK4/6 pathways can prevent RB phosphorylation and arrest cell cycle. Source data are provided as a Source Data file.

superior to chemotherapy + trastuzumab (Hazard ratio = 0.67; $p = 0.025$; 8.3 vs. 5.7 months) most likely due to the luminal tumors deriving larger benefit form CDK4/6 inhibition and endocrine therapy compared to chemotherapy.

Our study has two limitations worth noting. First, although a switch from HER2-E to Luminal A has been described after combinations of trastuzumab, pertuzumab and chemotherapy in clinical samples[13,37], we have not explored the molecular effects in vitro of dual HER2 blockade with trastuzumab and pertuzumab, as our in vitro models cannot recapitulate antibody-dependent cellular cytotoxicity. Second, although our preclinical data suggest that a switch to Luminal A could be leveraged with CDK4/6 inhibition, we do not have clinical evidence that patients with HER2-E tumors that become Luminal A upon anti-HER2 therapy benefit from CDK4/6 inhibitors. Further clinical studies are needed to address this preclinical hypothesis.

In conclusion, our findings support the use of maintenance anti-HER2 treatment in HER2+ breast cancer sensitive to anti-HER2 therapies, and warrant further research into exploiting molecular subtypes changes (i.e., HER2-E to Luminal) to improve patient outcomes.

## Methods

**PAMELA study.** The main results of the PAMELA neoadjuvant phase II study have been previously reported[7]. This study is registered with ClinicalTrials.gov, number NCT01973660, and it is completed. The study protocol was approved by independent ethics committees at each center. In this trial, 151 patients with early HER2+ breast cancer were treated with neoadjuvant lapatinib (1000 mg daily) and trastuzumab (8 mg kg$^{-1}$ i.v. loading dose followed by 6 mg kg$^{-1}$) for 18 weeks. Patients with HR+ breast cancer received letrozole or tamoxifen according to menopausal status. Tumor samples were collected at baseline, day 14 and surgery and subsequently formalin-fixed paraffin-embedded (FFPE). As previously reported[7], Ki67 was evaluated by IHC on FFPE tissue sections from core biopsies at baseline and at day 14 of treatment. These data are provided as a Source Data file.

**LPT109096 study.** The main results of the LPT109096 neoadjuvant phase II study have been previously reported[23]. This study is registered with ClinicalTrials.gov, number NCT00524303. In this trial, 100 patients with early HER2+ breast cancer were randomized to lapatinib, trastuzumab or both. Patients received 2 weeks of treatment without chemotherapy, and after a tumor biopsy, they received the same anti-HER2 treatment in combination with chemotherapy consisting of 5FU 500 mg m$^{-2}$ + epirubicin 75 mg m$^{-2}$ + cyclophosphamide 500 mg m$^{-2}$ i.v. every 21 days (FEC75) for four cycles followed by weekly paclitaxel 80 mg m$^{-2}$ for 12 weeks. Trastuzumab was administered every week (4 mg kg$^{-1}$ i.v. loading dose followed by 2 mg kg$^{-1}$). Lapatinib was administered every day (1250 mg if given without chemotherapy, 750 mg during FEC75 and 1000 mg during paclitaxel). Tumor samples were collected at baseline and day 14 and subsequently FFPE according to protocol.

**Cell lines and drugs.** The breast cancer cell lines BT474, SKBR3 and MCF7 were purchased from the American Type Culture Collection (ATCC, Manassas, VA, USA). All cell lines were maintained as recommended by the suppliers. All cell lines were authenticated using Human 9-Marker STR Profile and Interspecies Contamination Test by IDEXX BioAnalytics. Lapatinib, neratinib, tucatinib and palbociclib were purchased from Selleckchem (Houston, TX, USA). Trastuzumab was obtained from Roche Farma (Basel, Switzerland). A BT474-derived lapatinib and trastuzumab resistant (BT474-L$^R$T$^R$) cell line and a BT474-derived tucatinib and trastuzumab resistant (BT474-Tu$^R$T$^R$) cell line were established by treating BT474 with the corresponding TKI plus trastuzumab for 6 months, starting at low concentrations and increasing them at each passage. BT474-L$^R$T$^R$ are maintained with 2 nM lapatinib + 2 µg ml$^{-1}$ trastuzumab, while BT474-Tu$^R$T$^R$ are maintained with 2 nM tucatinib + 2 µg$^{-1}$ ml trastuzumab.

**RNA extractions.** RNA samples of the PAMELA trial from baseline and day 14 of treatment were previously extracted as reported[7]. Here, we extracted RNA from 57 residual surgical tumor FFPE material (non-pCR) from the PAMELA trial and 36 pairs of FFPE tumor samples (baseline and day 14) from the LPT109096 study using the High Pure FFPET RNA isolation kit (Roche, Indianapolis, IN, USA) following manufacturer's protocol. At least 1–5 10 µm FFPE slides were used for each tumor specimen, and macrodissection was performed to avoid contamination with normal breast tissue if needed. The RNeasy Mini Kit (Qiagen, Hilden, Germany) was used to extract RNA from cell lines plated in triplicates in 6-well plates at a density of 150,000 cells per well. RNA was quantified at the NanoDrop spectrophotometer (Thermo Fisher Scientific, Waltham, MA, USA).

**Gene expression analysis.** RNA samples of the PAMELA trial from HER2-E tumors at baseline ($n = 101$) and day 14 of treatment ($n = 96$) were previously analyzed[7]. Here, the nCounter platform (NanoString Technologies, Seattle, Washington, USA) analyzed RNA samples from 57 residual tumors from the PAMELA trial, RNA from 36 paired tumors from the LPT109096 study, and RNA from cell lines treated with anti-HER2 and anti-CDK4/6 therapies. A minimum of 100 ng of total RNA was used to measure the expression of 50 genes of the PAM50 intrinsic subtype predictor assay and 5 housekeeping genes (ACTB, MRPL19, PSMC4, RPLP0 and SF3A1). Expression counts were then normalized using the nSolver 4.0 software and custom scripts in R 3.4.3. For each sample we calculated the PAM50 signature scores (Basal-like, HER2-E, Luminal A and B, normal-like) and the proliferation signature score[38]. These data are provided as a Source Data file.

**Cell viability assays.** Cells were plated in triplicate at 4000 cells per well in 96-well plates. The day after seeding, cells were treated with increasing concentrations of TKI (lapatinib, neratinib and tucatinib) and 10 µg ml$^{-1}$ trastuzumab. Cell viability was determined 72 h after treatment using CellTiter 96 AQueous One Solution Cell Proliferation Assay (MTS) (Promega Corporation, Madison, Wisconsin, USA) following the manufacturer's instructions, and quantified using the Gen5 Microplate Reader and Imager Software (BioTek, Winooski, Vermont, USA). For anti-CDK4/6 treatments, cells were plated in triplicate at 4000 cells per well in 96-well plates and treated at low doses of TKI or DMSO for 24 h. Cells were then treated with increasing concentrations palbociclib in combination with low doses of TKI or DMSO. At 72 h, cells were labeled with the DNA stain Hoechst 33342 (Invitrogen, Life Technologies, Paisley, UK) and fluorescence was subsequently determined at the Gen5 Microplate Reader and Imager Software. GraphPad Prism was used for statistics. These data are provided as a Source Data file.

**Western blotting.** Cells were plated in triplicate at 150,000 cells per well in 6-well plates and treated with TKI and trastuzumab. Cell lysates were collected after 24 h of treatment with Pierce RIPA buffer (Thermo Fisher Scientific, Waltham, MA, USA) supplemented with protease inhibitors: 5 µM NaF, 1 µM PMSF, 1 µM Na$_3$O$_4$V, 1 µM benzamidine, 1 µg ml$^{-1}$ aprotinin, 1 µM leupeptin, 1 µM DTT. Total protein extracts were quantified using the DC Protein Assay (BioRad Laboratories, Hercules, California, USA). 25 µg of proteins were separated in reducing conditions (2.5% β-mercaptoethanol) by SDS–PAGE and transferred to nitrocellulose membranes (BioRad Laboratories, Hercules, California, USA) for further processing, following standard Western blotting procedures. Primary antibodies used in this study were: HER2 (D8F12), Phospho-HER2 (Tyr1221/1222) (6B12), AKT, Phospho-Akt (Ser473) (D9E), Phospho-RB (Ser807/811) (D20B12), Cyclin D1 (92G2) and anti-GAPDH (14C10) rabbit antibodies from Cell Signaling Technologies (Massachusetts, USA). The secondary fluorescent antibody used was the IRDye 800CW Donkey anti-Rabbit IgG (LI-COR Biosciences, Lincoln, Nebraska, USA). Fluorescent signal was acquired by the Odyssey Imaging System (LI-COR Biosciences, Lincoln, Nebraska, USA). Uncropped blots are provided as a Source Data file.

**Cell cycle analysis.** BC cell lines were fixed in 70% cold ethanol after 72 h of treatment with combinations of 10 µg ml$^{-1}$ trastuzumab and TKI (10 nM lapatinib, 2 nM neratinib or 10 nM tucatinib) or with DMSO control. Propidium Iodide (PI) was used to stain total DNA. Data acquisition was performed at the Cytometry and cell sorting core facilities of the Institut d'Investigacions Biomèdiques August Pi i Sunyer (IDIBAPS) using a BD FacsDiva analyzer. GraphPad Prism was used for statistics. These data are provided as a Source Data file.

**Clonogenic assay.** Breast cancer cell lines were plated at low density in 6-well plates and treated with combinations of 10 µg ml$^{-1}$ trastuzumab and TKI (10 nM lapatinib, 2 nM neratinib or 10 nM tucatinib) or with DMSO control. After 10 days, cells were fixed with cold methanol and stained with 0.5% crystal violet solution.

**Statistical analysis.** To identify genes and signatures whose expression was significantly different between paired samples (baseline vs day 14 or day 14 vs. surgery for clinical samples, or untreated vs treated for cell lines) or unpaired samples (parental cell lines vs anti-HER2 resistant cell lines) we used two-class paired Significance Analysis of Microarrays (SAM) with a false discovery rate (FDR) <5%. All statistical tests were two sided, and the statistical significance level was set to <0.05.

**Reporting Summary.** Further information on research design is available in the Nature Research Reporting Summary linked to this article.

## Data availability
The source data underlying Figs. 1b, c, 2a–d, 3b–d, 4b, 5a, and Supplementary Figs. 1c, 3a–d, 4a and 6 are provided as a Source Data file, and are also available from the corresponding author upon reasonable request.

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

## Acknowledgements

This study was funded by the project P116/00904, integrated in the Plan Estatal I+D+I and co-funded by Instituto de Salud Carlos III - Subdirección General de Evaluación and European Regional Development Fund (ERDF) (to A.P.), Pas a Pas (to A.P.), Save the Mama (to A.P.), Breast Cancer Now - 2018NOVPCC1294 (to A.P.), and Fundación Científica Asociación Española Contra el Cáncer - Ayuda Postdoctoral AECC 2017 (to F. B-M). We are indebted to the Biobank and the Cytometry and cell sorting core facilities of the Institut d'Investigacions Biomèdiques August Pi i Sunyer (IDIBAPS) for their technical help.

## Author contributions

Experimental study design: F.B-M. and A.P. Data acquisition: F.B-M., G.G., T.P., L.P., P.N., A.L.-C., B.B., M.O., S.M., N.M., M.V., B.A., O.M., S.P., R.L., M.M., N.C., P.G., I.G., L.M., J.A., E.M., S.G., R.R.G., P.V., J.C., E.C. and A.P. Data analysis: F.B-M., G.G., T.P., L.P., P.N., P.G and A.P. Data interpretation: F.B-M., G.G., T.P., L.P., P.N., A.L.-C., B.B., M.O., S.M., N.M., M.V., B.A., O.M., S.P., R.L., M.M., N.C., P.G., I.G., L.M., J.A., E.M., S.G., R.R.G., P.V., J.C., E.C. and A.P. Writing of the manuscript: F.B-M. and A.P. Review of the manuscript: all authors.

## Competing interests

A.P. reports consulting fees from Nanostring Technologies Roche, Pfizer, Novartis and Daiichi Sankyo outside the submitted work. G.G. reports travel support from Pfizer. M. O. reports grant support (to the Institution) from AstraZeneca, Philips Healthcare, Genentech, Roche, Novartis, Immunomedics, Seattle Genetics, GSK, Boehringer-Ingelheim, PUMA Biotechnology, consultant fees from Roche, GSK, PUMA Biotechnology and honoraria from Roche, Novartis and travel grants from Roche, Pierre-Fabre, Novartis, GP Pharma, Grünenthal outside the submitted work. S.P. has received honoraria for travel grants from Roche and has served as an advisor/consultant to Polyphor outside of the submitted work. N.M. is a speaker honorarium from AstraZeneca, Roche, Novartis, Celgene, Eisai, and Pfizer outside the submitted work. E.C. reports personal fees from Roche, Lilly, Pfizer and Novartis outside the submitted work. The remaining authors declare no competing interests.
