## [Peer Review File · Nature Communications]

Reviewers' comments:

Reviewer #1 (Remarks to the Author):

General Comment:

Introduction: I find it interesting that it is considered surprising that not all patients with HER2-E cancers respond to HER2 targeted therapy. Given that, as the author's state, the gene expression profile that results in classification into the HER2-E subtype can result from HER2 gene amplification/overexpression, but also from amplification of other receptor tyrosine kinase oncogenes such as FGFR, one should not expect that patients with HER2-E expression profiles but without HER2 gene amplification/overexpression should respond to these treatments. The principle of Oncogene Addiction is what drives the response to HER2 targeted drugs, and absent that, regardless of the gene expression profile, there should be no expectation of response or treatment to HER2 targeted drugs.

Results:

1. On page 3 the authors state: "Although similar biological changes were 3 observed in HER2+/HR+/HER2-E and HER2+/HR-negative/HER2-E diseases, a PAM50 subtype switch from HER2-E to Luminal A was observed in 31.6% of HR+ tumors and 4.8% of HR-negative tumors". I am confused by the last part of this sentence. Looking at figure 1b, it looks like the vast majority of HER2+/HR-/HER2-E patients showed increase in luminal A signature. I don't see how figure 1c shows that only 4.8% of patients showed this change.
2. Also on page 3, the authors write: "Finally, we evaluated gene expression changes at day 14 in HER2+/HR+ tumors that were Luminal A (n=22) or B (n=16) at baseline. This gene expression profile was strongly correlated (correlation coefficient=0.89) with the gene expression profile obtained from HER2+/HR+/HER2-E disease, suggesting that anti-HER2 therapies have a higher impact on molecular tumor changes at day 14 than endocrine therapy (Supplementary Figure 1d)." I don't understand this sentence at all. How does this observation suggest that HER2 therapies have a higher impact on molecular changes than hormonal therapies? If they were luminal A or B at baseline, why would they receive HER2 targeted therapies to begin with? This is very confusing.
3. Also on page 3, the authors write: "As expected, all anti-HER2 treatments evaluated induced cell death in BT474 and SKBR3 but not in MCF7 (Fig. 2a)." This is simply incorrect and must be changed. Indeed, this mis-interpretation of these data is related to mis-interpretation of other data later in the paper. The "cell viability" assay used to generate these data cannot distinguish between the effects of a targeted agent on cell proliferation versus cell death. It is well known that these agents have a profound impact on cell proliferation, and this can explain the results shown in figure 2. And, indeed, there are published data that shows that short term treatment with HER2 targeted drugs does induce complete cell cycle arrest that is completely reversible, because there is little or no effect on clonogenic potential and cell viability.
4. On page 4 the authors write: "Regarding PAM50 subtype changes, a switch to Luminal A following dual HER2 blockade was observed in BT474 (HER2+/HR+/HER2-E) but not in SKBR3 (HER2+/HR-negative/HER2-E) (Fig. 2d)." That is not how I read figure 2D. While the increase in the luminal A subtype in SKBR3 cells is not as robust as for BT474, most of the red lines in that figure are increasing post-treatment, and you indicate a p-value of 0.005, so that is not consistent with the statement.
5. Next the author's show both in patient samples and in cell lines, that the effects of HER2 targeted drugs on gene expression profiles are reversible once the treatment is stopped. If one understands that most of the effects of the drugs is on cell cycle arrest and not cell death, then this reversal is exactly what one would predict. Indeed, there is good published data on the effects of HER inhibitors on gene expression in the setting of gene amplification, and any cells that survive such treatment would be expected to revert to the original once HER2 inhibition is discontinued. I don't think this has anything to do with acquired resistance to HER2 targeted drugs.
6. On page 5 and 6, the authors discuss the acquisition of palbociclib sensitivity in BT474 cells after treatment with HER2 targeted drugs. The data from this experiment are interesting, but the figure is confusing. Please separate the data derived from the resistant cells from the parental

controls to make it easier to see the effects of palbo in HER2-drug treated cells. Also, given the comment above regarding SKBR3 cells, why wasn't this experiment done, or reported for SKBR3 cells? This should be shown along with the BT474 data.

In summary, this is an interesting paper that documents transient changes in gene expression in HER2+ breast cancers following treatment with HER2 targeted drugs. The most interesting and potential important observations related to the switch to the luminal A expression pattern that may result in new sensitivity to palbociclib. That said, the data presented in this paper regarding that important point seem preliminary and need to be fleshed out further with many HER2+ and HER2-E cells lacking HER2 gene amplification.

Reviewer #2 (Remarks to the Author):

Review of paper by Braso-Maristany et al NCOMMS-19-21254

After the approval of trastuzumab as the first treatment for breast cancer patients with ERBB2 amplification, diagnostics to pick out this group was often based on HER2 levels by IHC. Many, but not all patients, referred to as HER2-positive, did respond to antibody treatment. In the past decade gene expression profiling, which was pioneered by the Perou group, including Prat the last author of the current manuscript, allowed a further refinement of breast cancer sub-types, describing 5 molecular sub-types. One of these referred to as HER2-enriched (HER2-E), turned out to be quite heterogeneous, containing not only the HER2-positive group with the amplicon, but HER2 overexpressors that also had expression profiles similar to the other molecular subtypes: Luminal A, luminal B, basal-like and normal-like. The group of HER2-E patients is also treated with HER2 targeting agents including trastuzumab, in combination with different ErbB family kinase inhibitors that have a different MoA. Unfortunately, not all respond as well as expected to these combination treatments.

With that as a basis, the main goal of the work presented in the manuscript NCOMMS-19-21254 by Braso-Maristany and colleagues is to provide more molecular understanding of the biology that underlies the response/lack of response to HER2-directed therapies, in the context of HER2+/HER2-E disease, with or without hormone receptor (HR).

What is most exciting and novel about this work is their usage of breast cancer samples from important clinical trials, PAMELA and LPT109096 phase II trials. In the PAMELA trial patients received trastuzumab plus lapatinib, a EGFR/HER2 kinase inhibitor, plus or minus hormone therapy dependent upon HR status, in the neoadjuvant setting. They had access to 96 tumors sample before and after 2 weeks of treatment. For the LPT109096 trial in which patients were treated individually with trastuzumab, lapatinib or the combination, they had access to 36 patients and material was collected at the start and after 2 weeks. For this trial they had smaller sample numbers, but they were able to back up at least some of the data gleaned from the PAMELA trial material.

For all the samples the expression of the PAM-50 genes, 6 intrinsic signatures defining the subtype and an 11-gene proliferation score were assessed. The most important result from the PAMELA analysis was the finding that in the HER2+/HR+/HER2-E group there was a subtype switch into luminal A in a high % of patients, and a proliferation decrease. Comparing the PAMELA results with those from the LPT109096 trial showed a high degree of similarity of changes in the expression profiles, but intrinsic subtype switching in the HER2+/HR+/HER2-E to luminal A following combination treatment was not observed. I would agree with their suggestion that this is probably due to the small sample size is true. I think it is appropriate and important to keep this analysis in the paper. In this first section, the data were well presented and backed up the conclusions.

In the second set of experiments, they turn to HER2+ breast cancer cell lines, either ER+ or ER-

and a HER2-/ER+ cell line. PAM50 subtyping was done, as well as cell death assays and western analyses of the cells treated with trastuzumab combined with lapatinib or 2 other TKIs with different ErbB selectivity. As expected, only the HER2+ cell lines and not the ER+ MCF7 cells were killed by the inhibitors. As seen for the clinical samples, HER2 blockade induced an increase in Luminal A and normal genes and a decrease in the other signatures. These results nicely back up the human results.

The next set of experiments examined samples of PAMELA patients after the complete 18 weeks of neoadjuvant therapy and before surgery of residual tumor material that was seen in 57 of the 96 patients. Gene expression profiling reveals many changes including up-regulation of proliferation related genes, an expected result, but very important to document. They also note that the luminal A signature observed in many at 2 weeks reverted before surgery and at least for some cases the HER2 signature appeared (Fig 3b).

Question to clarify - It was not clear to me if there was a correlation between the length of time after the last treatment (at 18wks) and the surgery and the loss of the Luminal A signature. Would they predict that surgery, even if there is no obvious disease, should be done right after treatment is stopped?

Next- they returned to the HER2+ model cell lines and waited to see changes 72 hrs after the last treatment.

Clarification - They mention that all the PAM50 genes rebounded after stopping therapy (Supp data 5). I noted that 100% of the genes rebounded in the SKBR3 model but only 75% did for the BT474 model. Please clarify.

They also generated resistant BT474 models after growing cells for long periods in Ab+TKI combinations. None of the resistant models switch to the Luminal A phenotype following inhibitor treatment, nor do they die.

Finally, they test whether or not the parental BT474 HER2 inhibitor treated cells that had a luminal A phenotype would become sensitive to the CDK 4/6 inhibitor palbocicib that is used for ER+ luminal disease. BT474 cells are somewhat sensitive to the CDK inhibitor alone (black line Fig 5a), excitingly, however, in the presence of the Ab or the 3 TKIs, there is enhanced sensitivity to palbocicib. In contrast, the resistant cells do not get resensitized to HER2 targeted drugs in the presence of the CDK inhibitor. These results are very interesting and suggest that the model that they propose in panel b - namely that after dual HER2 blockade the treatment should continue with a CDK4/6 inhibitor- should be tested in the clinic.

Suggestion/question - what happens to the BT474 cells if the CDK inhibitor is added to the dual treatment, which might be more reflective of the clinical setting.

Minor correction - On pg 4 they refer to Supp Fig 4a and 3b. I believe that this should read 4b.

Nancy Hynes

Dear Reviewers,

Thank you very much for your relevant comments and constructive criticism. We believe our manuscript NCOMMS-19-21254, "*Phenotypic changes of HER2-positive breast cancer during and after dual HER2 blockade*" has improved substantially after revision and we hope that you now find it acceptable for publication.

Please find below a point-by-point response to your comments.

Thank you very much for your time and consideration,

Aleix Prat, MD PhD

Head Medical Oncology, Hospital Clínic Link
Associate Professor, University of Barcelona (UB)
Director, Breast Pathology - Senology Master's degree, UB. Link
Head of Translational Genomics and Targeted Therapeutics Group, IDIBAPS. Link

Address (google maps):

Hospital Clínic
Villarroel 170, Escalera 2, Planta 5
08036, Barcelona, Spain
alprat@clinic.cat

Reviewers' comments:

Reviewer #1 (Remarks to the Author):

General Comment:

Introduction: I find it interesting that it is considered surprising that not all patients with HER2-E cancers respond to HER2 targeted therapy. Given that, as the author's state, the gene expression profile that results in classification into the HER2-E subtype can result from HER2 gene amplification/overexpression, but also from amplification of other receptor tyrosine kinase oncogenes such as FGFR, one should not expect that patients with HER2-E expression profiles but without HER2 gene amplification/overexpression should respond to these treatments. The principle of Oncogene Addiction is what drives the response to HER2 targeted drugs, and absent that, regardless of the gene expression profile, there should be no expectation of response or treatment to HER2 targeted drugs.

Results:

1. On page 3 the authors state: "Although similar biological changes were 3 observed in HER2+/HR+/HER2-E and HER2+/HR-negative/HER2-E diseases, a PAM50 subtype switch from HER2-E to Luminal A was observed in 31.6% of HR+ tumors and 4.8% of HR-negative tumors". I am confused by the last part of this sentence. Looking at figure 1b, it looks like the vast majority of HER2+/HR-/HER2-E patients

showed increase in luminal A signature. I don't see how figure 1c shows that only 4.8% of patients showed this change.

We thank the reviewer for this comment. We would like to clarify the difference between an increase or decrease of a PAM50 signature score versus a real switch of a PAM50 molecular subtype. The PAM50 algorithm consists of a centroid-based prediction method that uses the expression of a set of 50 genes to assess 5 signature scores (i.e. Luminal A, Luminal B, HER2-enriched [HER2-E], Basal-like, Normal-like). For each sample, the PAM50 algorithm gives 5 signatures/centroids scores, and assigns a subtype to a sample based on the signature score with the highest absolute value¹. Thus, a treated sample can have a significant increase in the Luminal A signature but still not being called Luminal A by PAM50 since the Luminal A signature score might not be the highest. This is exactly what happens with SKBR3 cell line. Anti-HER2 therapy induces a relative change in the Luminal A signature, as shown in Fig. 1c; however, the HER2-E signature is still the one with the highest score; thus, the PAM50 algorithm calls SKBR3 HER2-E despite an increase in the Luminal A signature during anti-HER2 treatment.

2. Also on page 3, the authors write: “Finally, we evaluated gene expression changes at day 14 in HER2+/HR+ tumors that were Luminal A (n=22) or B (n=16) at baseline. This gene expression profile was strongly correlated (correlation coefficient=0.89) with the gene expression profile obtained from HER2+/HR+/HER2-E disease, suggesting that anti-HER2 therapies have a higher impact on molecular tumor changes at day 14 than endocrine therapy (Supplementary Figure 1d).” I don't understand this sentence at all. How does this observation suggest that HER2 therapies have a higher impact on molecular changes than hormonal therapies? If they were luminal A or B at baseline, why would they receive HER2 targeted therapies to begin with? This is very confusing.

We thank the reviewer for this comment. First, we would like to clarify that in the PAMELA and LPT109096 trials, all patients had HER2+ breast tumors and all patients received anti-HER2 therapies. Second, patients with HER2+/HR+ tumors in the PAMELA trial were also treated with endocrine therapies². Third, within HER2+/HR+ disease, approximately 50% of tumors are Luminal A and Luminal B by PAM50 and the other 50% are HER2-E³. Thus, a major question that was raised was how much the changes in gene expression at day14 in patients with HER2+/HR+ disease was due to endocrine therapy. To try address this, we compared the changes in gene expression in HER2+/HR+/Luminal A/B disease with HER2+/HR+/HER2-E disease. Overall, no clearly differences were observed, suggesting that the main driver of the gene expression changes was the anti-HER2 therapy and not the endocrine treatment.

In order to clarify the points raised by the reviewer, we have changed the text (page 3): *“Finally, since patients with HER2+/HR+ disease in the PAMELA trial received anti-*

HER2 therapy in combination with letrozole or tamoxifen, and treatment with letrozole alone for 2 weeks has shown to reduce Ki67 in HER2+/HR+ tumors in the PER-ELISA phase II trial, specially within Luminal A and B disease⁴, we separated and compared the gene expression changes at day 14 in the PAMELA trial of HER2+/HR+/Luminal A or B tumors (n=38) with HER2+/HR+/HER2-E tumors (n=38). The analysis revealed a strong correlation (correlation coefficient=0.89) between both gene lists, suggesting that anti-HER2 therapies have a higher impact on molecular tumor changes at day 14 than endocrine therapy (Supplementary Figure 1d), although we cannot exclude an additive or a synergistic effect between anti-HER2 therapy and endocrine therapy in this group of patients.”

3. Also on page 3, the authors write: “As expected, all anti-HER2 treatments evaluated induced cell death in BT474 and SKBR3 but not in MCF7 (Fig. 2a).” This is simply incorrect and must be changed. Indeed, this mis-interpretation of these data is related to mis-interpretation of other data later in the paper. The “cell viability” assay used to generate these data cannot distinguish between the effects of a targeted agent on cell proliferation versus cell death. It is well known that these agents have a profound impact on cell proliferation, and this can explain the results shown in figure 2. And, indeed, there are published data that shows that short term treatment with HER2 targeted drugs does induce complete cell cycle arrest that is completely reversible, because there is little or no effect on clonogenic potential and cell viability.

We thank the reviewer for this important comment. We agree and have now changed the text accordingly. Moreover, we agree that the observed effects on cell viability can be due to cell-cycle arrest caused by HER2 inhibition. Therefore, we have performed cell cycle experiments and confirmed that anti-HER2 treatments in BT474 and SKBR3 cause cell cycle arrest at G1. Furthermore, we now show that the clonogenic potential is also impaired by dual blockade treatment.

These data have now been included in the manuscript (page 4). *“As expected, all anti-HER2 treatments evaluated reduced cell viability in a dose-dependent manner in BT474 and SKBR3 (Fig. 2a and Supplementary Figure 3b-c) but not in HCC1954, which has a mutation in PIK3CA that might confer resistance to anti-HER2 therapy⁵, or MCF7, which does not overexpress HER2 (Supplementary Figure 3d). Concordant with this observation, phosphorylation of HER2 and phosphorylation of the survival kinase AKT were reduced upon treatment in BT474 and SKBR3, demonstrating successful pathway inhibition (Fig. 2b). Dual HER2 blockade arrested cell cycle at G1 (Supplementary Figure 4a) and reduced clonogenic potential (Supplementary Figure 4b), consistent with known roles of HER2 downstream signaling pathways⁶”.*

4. On page 4 the authors write: “Regarding PAM50 subtype changes, a switch to Luminal A following dual HER2 blockade was observed in BT474

(HER2+/HR+/HER2-E) but not in SKBR3 (HER2+/HR-negative/HER2-E) (Fig. 2d).” That is not how I read figure 2D. While the increase in the luminal A subtype in SKBR3 cells is not as robust as for BT474, most of the red lines in that figure are increasing post-treatment, and you indicate a p-value of 0.005, so that is not consistent with the statement.

We do agree with the reviewer that the Luminal A signature increases significantly upon dual HER2 blockade in the HER2+/HR-negative/HER2-E cell line SKBR3. However, this change of the signature score is not enough to induce a change in the molecular subtype. In BT474, however, the increase in the Luminal A signature score leads to a subtype change since the other PAM50 signatures have lower absolute scores. Therefore, the 2 observations are related but different. A true subtype switch requires more than just an increase in a signature score.

5. Next the author’s show both in patient samples and in cell lines, that the effects of HER2 targeted drugs on gene expression profiles are reversible once the treatment is stopped. If one understands that most of the effects of the drugs is on cell cycle arrest and not cell death, than this reversal is exactly what one would predict. Indeed, there is good published data on the effects of HER inhibitors on gene expression in the setting of gene amplification, and any cells that survive such treatment would be expected to revert to the original once HER2 inhibition is discontinued. I don’t think this has anything to do with acquired resistance to HER2 targeted drugs.

We thank the reviewer for this comment. In patients with HER2+/HER2-E disease, anti-HER2 treatment can induce 100% cell death and apoptosis since a substantial proportion of patients achieve a pCR at surgery after neoadjuvant anti-HER2-based treatment. This proportion is 40% if dual anti-HER2 treatment is given without chemotherapy and 80% if dual HER2 blockade is combined with chemotherapy. Said that, we agree (and also show now in page 4) that most cells that do not die during anti-HER2 treatment are still sensitive to the treatment and show cell-cycle arrest. However, these tumors or cell lines that survive during anti-HER2 therapy eventually become anti-HER2 resistant and have a poor outcome. For example, patients with HER2+/HER2-E disease that do not achieve a pCR at surgery have a very poor survival despite treatment with adjuvant anti-HER2 therapy. In addition, BT474 cells after long periods of anti-HER2 therapy become resistant and re-gain the HER2-E phenotype. In this scenario is where we feel acquired resistance has occurred.

6. On page 5 and 6, the authors discuss the acquisition of palbociclib sensitivity in BT474 cells after treatment with HER2 targeted drugs. The data from this experiment are interesting, but the figure is confusing. Please separate the data derived from the resistant cells from the parental controls to make it easier to see the effects of palbo in HER2-drug treated cells. Also, given the comment above regarding SKBR3 cells, why wasn’t this experiment done, or reported for SKBR3 cells? This should be shown

along with the BT474 data. In summary, this is an interesting paper that documents transient changes in gene expression in HER2+ breast cancers following treatment with HER2 targeted drugs. The most interesting and potential important observations related to the switch to the luminal A expression pattern that may result in new sensitivity to palbociclib. That said, the data presented in this paper regarding that important point seem preliminary and need to be fleshed out further with many HER2+ and HER2-E cells lacking HER2 gene amplification.

As suggested, we have extended the cell line models treated with anti-HER2 and palbociclib. In total we have used 5 cell lines: BT474, SKBR3, HCC1954 and the two BT474-derived anti-HER2-resistant cells (**Fig. 5a**). HER2 blockade sensitized BT474 and SKBR3 cells to palbociclib, while HCC1954, which has a *PIK3CA* mutation and is intrinsically resistant to anti-HER2 therapies, did not benefit from the combination. Finally, BT474-L^RT^R and BT474-Tu^RT^R cell lines were insensitive to palbociclib, further demonstrating that the HER2-E phenotype confers resistance to CDK4/6 inhibitors. Overall, these data suggested that anti-HER2 treatment can modulate the sensitivity to CDK4/6 inhibition in HER2-E disease by inducing a more Luminal A-like phenotype, while cells that are resistant to anti-HER2 treatments remain HER2-E and do not benefit from the combination of anti-HER2 therapies and CDK4/6 inhibition. These data have been added to the manuscript (page 6).

Reviewer #2 (Remarks to the Author):

Review of paper by Braso-Maristany et al NCOMMS-19-21254

After the approval of trastuzumab as the first treatment for breast cancer patients with ERBB2 amplification, diagnostics to pick out this group was often based on HER2 levels by IHC. Many, but not all patients, referred to as HER2-positive, did respond to antibody treatment. In the past decade gene expression profiling, which was pioneered by the Perou group, including Prat the last author of the current manuscript, allowed a further refinement of breast cancer sub-types, describing 5 molecular sub-types. One of these referred to as HER2-enriched (HER2-E), turned out to be quite heterogenous, containing not only the HER2-positive group with the amplicon, but HER2 overexpressors that also had expression profiles similar to the other molecular subtypes: Luminal A, luminal B, basal-like and normal-like. The group of HER2-E patients is also treated with HER2 targeting agents including trastuzumab, in combination with different ErbB family kinase inhibitors that have a different MoA.

Unfortunately, not all respond as well as expected to these combination treatments.

With that as a basis, the main goal of the work presented in the manuscript NCOMMS-19-21254 by Braso-Maristany and colleagues is to provide more molecular understanding of the biology that underlies the response/lack of response to HER2-directed therapies, in the context of HER2+/HER2-E disease, with or without hormone receptor (HR).

What is most exciting and novel about this work is their usage of breast cancer samples from important clinical trials, PAMELA and LPT109096 phase II trials. In the PAMELA trial patients received trastuzumab plus lapatinib, a EGFR/HER2 kinase inhibitor, plus or minus hormone therapy dependent upon HR status, in the neoadjuvant setting. They had access to 96 tumors sample before and after 2 weeks of treatment. For the LPT109096 trial in which patients were treated individually with trastuzumab, lapatinib or the combination, they had access to 36 patients and material was collected at the start and after 2 weeks. For this trial they had smaller sample numbers, but they were able to back up at least some of the data gleaned from the PAMELA trial material.

For all the samples the expression of the PAM-50 genes, 6 intrinsic signatures defining the sub-type and an 11-gene proliferation score were assessed. The most important result from the PAMELA analysis was the finding that in the HER2+/HR+/HER2-E group there was a subtype switch into luminal A in a high % of patients, and a proliferation decrease. Comparing the PAMELA results with those from the LPT109096 trial showed a high degree of similarity of changes in the

expression profiles, but intrinsic subtype switching in the HER2+/HR+/HER2-E to luminal A following combination treatment was not observed. I would agree with their suggestion that this is probably due to the small sample size is true. I think it is appropriate and important to keep this analysis in the paper. In this first section, the data were well presented and backed up the conclusions.

In the second set of experiments, they turn to HER2+ breast cancer cell lines, either ER+ or ER- and a HER2-/ER+ cell line. PAM50 subtyping was done, as well as cell death assays and western analyses of the cells treated with trastuzumab combined with lapatinib or 2 other TKIs with different ErbB selectivity. As expected, only the HER2+ cell lines and not the ER+ MCF7 cells were killed by the inhibitors. As seen for the clinical samples, HER2 blockade induced an increase in Luminal A and normal genes and a decrease in the other signatures. These results nicely back up the human results.

The next set of experiments examined samples of PAMELA patients after the complete 18 weeks of neoadjuvant therapy and before surgery of residual tumor material that was seen in 57 of the 96 patients. Gene expression profiling reveals many changes including up-regulation of proliferation related genes, an expected result, but very important to document. They also note that the luminal A signature observed in many at 2 weeks reverted before surgery and at least for some cases the HER2 signature appeared (Fig 3b).

Question to clarify - It was not clear to me if there was a correlation between the length of time after the last treatment (at 18wks) and the surgery and the loss of the Luminal A signature. Would they predict that surgery, even if there is no obvious disease, should be done right after treatment is stopped?

We thank the reviewer for this comment, which we have investigated in 57 patients with HER2+/HER2-E disease that did not achieve a pCR. The median time from the last dose of therapy to surgery, where the tumor samples were obtained, was 29.5 days (range=7-76 days). 31 patients had treatment stopped <1 month before surgery (median=25; range=7-30), while 39 patients underwent surgery >1 month after treatment was stopped, as reported in Source Data file. However, we have not found a correlation between the rebound effect and the length of time between the last dose of therapy and surgery, as depicted in the plots below:

We have added a sentence in the manuscript, please see page 5: *“However, we did not observe a correlation between the rebound effect and the number of days from the last dose and the day of the surgical procedure (data not shown)”*. Finally, regarding the question if surgery should be done right after stopping therapy, our results do not shed light on this.

Next- they returned to the HER2+ model cell lines and waited to see changes 72 hrs after the last treatment.

Clarification - They mention that all the PAM50 genes rebounded after stopping therapy (Supp data 5). I noted that 100% of the genes rebounded in the SKBR3 model but only 75% did for the BT474 model. Please clarify.

We thank the reviewer for noticing this error. We have now changed the text accordingly.

They also generated resistant BT474 models after growing cells for long periods in Ab+TKI combinations. None of the resistant models switch to the Luminal A phenotype following inhibitor treatment, nor do they die.

Finally, they test whether or not the parental BT474 HER2 inhibitor treated cells that had a luminal A phenotype would become sensitive to the CDK 4/6 inhibitor palbocicib that is used for ER+ luminal disease. BT474 cells are somewhat sensitive to the CDK inhibitor alone (black line Fig 5a), excitingly, however, in the presence of the Ab or the 3 TKIs, there is enhanced sensitivity to palbocicib. In contrast, the resistant cells do not get resensitized to HER2 targeted drugs in the presence of the CDK inhibitor. These results are very interesting and suggest that the model that they propose in panel b – namely that after dual HER2 blockade the treatment should continue with a CDK4/6 inhibitor- should be tested in the clinic.

Suggestion/question – what happens to the BT474 cells if the CDK inhibitor is added to the dual treatment, which might be more reflective of the clinical setting.

We thank the reviewer for this important suggestion and we agree that the triple combination is a clinically relevant question. However, BT474 and SKBR3 are extremely sensitive to dual HER2 inhibition. Although we have worked with several concentrations of

trastuzumab, we find that BT474 and SKBR3 do not respond to trastuzumab in a dose-dependent manner. Thus, we struggle to see an additive or synergistic effect of palbociclib in the presence of dual HER2 blockade.

Minor correction – On pg 4 they refer to Supp Fig 4a and 3b. I believe that this should read 4b.

The text has been corrected accordingly. Thank you very much.

References

1. Parker, J. S. *et al.* Supervised risk predictor of breast cancer based on intrinsic subtypes. *J. Clin. Oncol.* **27**, 1160–7 (2009).
2. Llombart-Cussac, A. *et al.* HER2-enriched subtype as a predictor of pathological complete response following trastuzumab and lapatinib without chemotherapy in early-stage HER2-positive breast cancer (PAMELA): an open-label, single-group, multicentre, phase 2 trial. *Lancet Oncol.* **18**, 545–554 (2017).
3. Prat, A., Pascual, T. & Adamo, B. Intrinsic molecular subtypes of HER2+ breast cancer. *Oncotarget* **8**, 73362–73363 (2017).
4. Guarneri, V. *et al.* De-escalated therapy for HR+/HER2+ breast cancer patients with Ki67 response after 2-week letrozole: results of the PerELISA neoadjuvant study. *Ann. Oncol.* (2019). doi:10.1093/annonc/mdz055
5. Chakrabarty, A. *et al.* Trastuzumab-resistant cells rely on a HER2-PI3K-FoxO-survivin axis and are sensitive to PI3K inhibitors. *Cancer Res.* **73**, 1190–1200 (2013).
6. Oh, D.-Y. & Bang, Y.-J. HER2-targeted therapies — a role beyond breast cancer. *Nat. Rev. Clin. Oncol.* 1–16 (2019). doi:10.1038/s41571-019-0268-3

REVIEWERS' COMMENTS:

Reviewer #1 (Remarks to the Author):

I am satisfied with the response to the reviews and recommend acceptance for publication.

Reviewer #2 (Remarks to the Author):

I am satisfied with the revision of the manuscript and have no further comments for the authors.